# Protection and Safety Evaluation of Live Constructions Derived from the Pgm^−^ and pPCP1^−^
*Yersinia pestis* Strain

**DOI:** 10.3390/vaccines8010095

**Published:** 2020-02-21

**Authors:** Xiuran Wang, Amit K. Singh, Wei Sun

**Affiliations:** Department of Immunology and Microbial Disease, Albany Medical College, Albany, NY 12208, USA; wangx9@amc.edu (X.W.); singha5@amc.edu (A.K.S.)

**Keywords:** *Yersinia pestis*, Pgm^−^, pPCP1^−^, lipid A, protection, safety

## Abstract

Based on a live attenuated *Yersinia pestis* KIM10(pCD1Ap) strain (Pgm^−^, pPCP1^−^), we attempted to engineer its lipid A species to achieve improvement of immunogenicity and safety. A mutant strain designated as YPS19(pCD1Ap), mainly synthesizing the hexa-acylated lipid A, and another mutant strain designated as YPS20(pCD1Ap), synthesizing 1-dephosphalated hexa-acylated lipid A (detoxified lipid A), presented relatively low virulence in comparison to KIM10(pCD1Ap) by intramuscular (i.m.) or subcutaneous (s.c.) administration. The i.m. administration with either the KIM10(pCD1Ap) or YPS19(pCD1Ap) strain afforded significant protection against bubonic and pneumonic plague compared to the s.c. administration, while administration with completely attenuated YPS20(pCD1Ap) strain failed to afford significant protection. Antibody analysis showed that i.m. administration induced balanced Th1 and Th2 responses but s.c. administration stimulated Th2-biased responses. Safety evaluation showed that YPS19(pCD1Ap) was relatively safer than its parent KIM10(pCD1Ap) in Hfe^−/−^ mice manifesting iron overload in tissues, which also did not impair its protection. Therefore, the immune activity of hexa-acylated lipid A can be harnessed for rationally designing bacteria-derived vaccines.

## 1. Introduction

The live EV76 vaccine is a spontaneous *pgm* mutant that has been used in humans for over 80 years in regions and countries of the former Soviet Union (FSU) without any deaths reported [1,2,3]. The EV76 vaccine conferred better protection against bubonic and pneumonic plague than killed vaccines in different animals, but it sometimes caused local and systemic reactions including fever, malaise, lymphadenopathy, erythema and large induration at the injection site [4,5,6,7], as well as disease in primates [6]. In addition, the live Pgm^−^ strain (*Y. pestis* KIM5) retained virulence by intranasal (i.n.) or intravenous (i.v.) administration [6,8,9,10], caused fatal septicemic plague in an individual with hereditary hemochromatosis manifesting as iron overload in tissues [11], and restored its virulence in hemojuvelin-knockout (*Hjv^−/−^*) mice mimicking human hereditary hemochromatosis [12]. Thus, variable virulence of the live vaccine strain in animals and humans has deterred this vaccine from gaining worldwide acceptance [13,14]. Nevertheless, the large amount of accumulated evidence suggests that the live Pgm^−^ vaccine strain is very close to becoming a human vaccine. The WHO 2018 plague workshop still listed the live attenuated *Y. pestis* vaccine as one of new-generation plague vaccines [15]. Thus, it is worthwhile to perfect this vaccine by resolving certain concerns.

One known strategy used by *Y. pestis* to evade host innate surveillance is to produce a tetra-acylated lipid A (the nonstimulatory form of LPS) that is not recognized by Toll-like receptor 4 (TLR4) in the mammalian host due to absence of LpxL (lauroyltransferase) [16,17,18,19]. Insertion of *E. coli lpxL* into the chromosome of the virulent *Y. pestis* KIM5+ strain (Pgm^+^) to generate strain χ10015(pCD1Ap) (Δ*lpxP*::P_lpxL_
*lpxL*) restored production of hexa-acylated lipid A at both 26 °C and 37 °C, and presented high attenuation by subcutaneous (s.c.) administration [20]. The s.c. immunization with χ10015(pCD1Ap) afforded significant protection against both bubonic and pneumonic plague challenge [20] and the further attenuated strain χ10030(pCD1Ap) (Δ*lpxP*::P_lpxL_
*lpxL* ΔP_crp_::TT *araC* P_BAD_
*crp*) was still lethal to mice (personal communication with Dr. Christopher K, Cote at CIV USAMRIID). In addition, we generated strain χ10027(pCD1Ap) (Δ*lpxP*::P_lpxL_
*lpxL* ∆*lacI*::P_lpp_
*lpxE*), which heterologously expresses the *lpxE* gene encoding the lipid A 1-phosphatase from *Francisella novicida* in *Y. pestis*, predominantly yielding 1-dephosphorylated hexa-acylated lipid A (monophosphoryl lipid A, MLPA) with significantly less stimulatory activity to several mammalian cells in vitro than the lipid A from χ10015 [21]. The lipid A dephosphorylation moderately decreased virulence of χ10027 in mice by s.c. or i.n. infection [21]. MPLA is an endotoxin derivative that has been approved by US and European authorities as a vaccine adjuvant in humans [22] and is 100- to 10,000-fold less toxic than native lipid A (biphosphoryl lipid A) [23,24]. Therefore, the live attenuated Pgm^−^ vaccine strain with corresponding lipid A modification may retain immunogenicity but display low virulence and reactogenicity. 

Another well-established virulence factor, the plasminogen-activating protease, Pla, encoded on the pPCP1 plasmid, was essential for dissemination of *Y. pestis* in the course of plague development [25,26,27]. Pla manipulated host cell death pathways to facilitate pulmonary infection by cleaving the Fas ligand [28], resulted in tissue destruction and hemorrhage in hosts infected with *Y. pestis* [25] and also thwarted T cell defense against the plague [29]. Removal of Pla clearly reduced virulence of *Y. pestis* [25,27,30]. A combination of deletion of the Braun lipoprotein gene (*lpp*) with curing of plasmid pPCP1 dramatically altered the virulence of *Y. pestis* CO92 in a mouse model of pneumonic plague [31]. Thus, the combination of the enhancement of immunogenic attributes by lipid A modification with attenuation by eliminating the pPCP1 plasmid on top of a Pgm^−^ pPCP1^−^
*Y. pestis* KIM-based strain might improve protective immunity and safety of the live *Y. pestis* vaccine. Here, we showed that high doses of the YPS20(pCD1Ap) strain (Δ*lpxP*::P_lpxL_
*lpxL* ∆*lacI*::P_lpp_
*lpxE* Pgm^−^ pPCP1^−^) by intramuscular (i.m.) or subcutaneous (s.c.) injection did not cause any mouse death compared with those of the KIM10(pCD1Ap) and YPS19(pCD1Ap) strain (Δ*lpxP*::P_lpxL_
*lpxL* Pgm^−^ pPCP1^−^). Measurement of protective immunity showed that mice subcutaneously administrated each mutant strain partially survived challenge with bubonic and pneumonic plague. While i.m. immunization with either the YPS19(pCD1Ap) or KIM10(pCD1Ap) strain provided significant protection in mice against bubonic and pneumonic plague without a substantial difference, i.m. immunization with YPS20(pCD1Ap) did not. Safety evaluation indicated that the YPS19(pCD1Ap) strain was relatively safer than KIM10(pCD1Ap) in *Hfe*^−/−^ mice.

## 2. Materials and Methods

### 2.1. Bacterial Strains, Plasmids, and Culture Conditions 

All bacterial strains and plasmids used in this study are listed in Table 1. *E. coli* strains were grown routinely at 37 °C in Luria–Bertani broth (LB) or LB solidified with 1.2% Bacto Agar (Difco). *Y. pestis* bacteria were grown routinely at 28 °C on heart infusion broth (HIB) plus Congo red agar plates and tryptose-blood agar (TBA) plates (Difco) [32]. Ampicillin (100 μg/mL, Amp), chloramphenicol (25 μg/mL, Cm) or 5% sucrose were supplemented as appropriate.

### 2.2. Mice

Animal care and experimental protocols were in accordance with the NIH “Guide for the Care and Use of the laboratory Animals” and were approved by the Institutional Animal Care and Use Committee at Albany Medical College (IACUC protocol # 17-02004). Swiss Webster outbred mice were purchased from Charles River Laboratories. *Hfe^−/−^* mice (B6.129S6-*Hfe^tm2Nca^*/J) [36] were purchased from The Jackson Laboratory. Wild-type (*Hfe*^+/+^) B6.129 mice as controls were purchased from Taconic. All deficient mice were bred at the animal facility of Albany Medical College. 

### 2.3. Construction of Y. pestis Mutant Strains

Construction of *Y. pestis* mutants with the Δ*lpxP*::P_lpxL_
*lpxL* or/and ∆*lacI*::P*lpp lpxE* insertion based on Y. pestis KIM10 (Pgm^−^ pPCP1^−^) was described in our previous report [21]. Briefly, the linear *lpxP-*U-*cat-sacB*-P_lpxL_::*lpxL*-*lpxP-*D fragment cut from plasmid pYA4578 was electroporated into KIM10 harboring pKD46. Electroporants were plated onto TBA + Cm (25 μg/mL of chloramphenicol) plates and incubated at 26 °C for 2~3 days. Integrants with insertion of the *lpxP-*U-*cat-sacB*-P_lpxL_::*lpxL*-*lpxP-*D DNA fragment into the correct chromosomal site were confirmed by PCR using a primer set LpxL1/Cm-V (Table 2). Mutant colonies were streaked onto TBA + Cm + 5% sucrose plates to isolate sucrose-sensitive colonies for the second-step recombination. Then, linear *lpxP-*U-P_lpxL_*lpxL*-*lpxP-*D DNA fragments cut from pYA4577 were electroporated into *lpxP-*U-*cat-sacB*-P_lpxL_
*lpxL*-*lpxP-*D KIM10(pKD46) intermediate cells. Colonies grown on TBA + 5% sucrose plates were confirmed to be sensitive to chloramphenicol, onto TBA + Cm plates, and then validated by PCR using a primer set LpxL1/LxpL2 (Table 2) to confirm that the P_lpxL_
*lpxL*-*lpxP* fragment was inserted into the correct chromosomal site. Finally, plasmid pKD46 was eliminated from Δ*lpxP*::P_lpxL_
*lpxL* KIM10(pKD46) to yield YPS19 (Δ*lpxP*::P_lpxL_
*lpxL*) (Table 1). Then, the ∆*lacI*::P_lpp_
*lpxE* mutation was introduced on top of YPS19 using the same procedures to generate YPS20 (Δ*lpxP*::P_lpxL_
*lpxL* ∆*lacI*::P_lpp_
*lpxE*) (Table 1). The correct clones were confirmed by PCR and DNA sequencing. 

### 2.4. Virulence Analysis in Mice

A single colony of each mutant strain was inoculated into HIB supplemented with ampicillin (100 μg/mL) for selection of plasmid pCD1Ap and grown overnight at 26 °C. Bacteria were diluted into 10 mL of fresh HIB enriched with 0.2% xylose and 2.5 mM CaCl_2_ to obtain an OD_620_ of 0.1 and then incubated at 26 °C for intramuscular (i.m.) or subcutaneous (s.c.) infection, or at 37 °C for intranasal (i.n.) infection, to reach an OD_620_ of 0.6. The cells were then harvested, and the pellet resuspended in 1 mL of isotonic PBS and then adjusted to an appropriate concentration. 

Groups of Swiss Webster mice (10/group, equal males and females) were administrated by i.m. or s.c. injection with 100 μL of bacterial suspension (10^7^ CFU) or by i.n. route with 40 μL of bacterial suspension (10^6^ CFU). Actual numbers of colony-forming units (CFU) inoculated were determined by plating serial dilutions onto TBA agar. 

### 2.5. Determination of Protective Efficacy

*Y. pestis* strains were grown as described above. Two groups of Swiss Webster mice (10/group, equal males and females) were immunized i.m. with 1 × 10^7^ CFU of a mutant strain in 100 μL of PBS on day 0 and boosted i.m. with 1 × 10^7^ CFU on day 21. A group of mice (10/group) were injected with 100 μL of PBS as a control. Blood was collected by sub-mandibular vein puncture at 2- and 4-weeks post immunization. At 42 days after initial immunization, animals anesthetized with a 1:5 xylazine/ketamine mixture were challenged intranasally with 5 × 10^3^ CFU of *Y. pestis* KIM6+(pCD1Ap) in 40 μL PBS. All infected animals were observed over a 15-day period. 

### 2.6. Immune Responses

ELISA was used to assay serum IgG antibodies against *Yersinia* whole cell lysates (YpL) of *Y. pestis* KIM5+ or purified rLcrV protein [37]. Polystyrene 96-well flat-bottom microtiter plates (Dynatech Laboratories Inc., Chantilly, VA, USA) were coated with 1 μg/well of YpL protein or rLcrV. The procedures were the same as those described previously [37]. 

### 2.7. Evaluation of Vaccine Safety

*Y. pestis* mutant strains were evaluated for safety using 7 wk old *Hfe^−/−^* mice (*n* = 10/group). Mice were administered intramuscularly with a single dose of 1 × 10^7^ CFU of each *Y. pestis* mutant strain. These mice were observed for signs of mortality and morbidity for 30 days. 

### 2.8. Statistical Analysis

The Graph-Pad Prism 8.0 was used to analyze data statistically. The log-rank (Mantel–Cox) test, and the two-way ANOVA were used for survival analysis, statistical analyses of spleen weight and cytokine analysis, respectively. Data were expressed as means ± standard deviation (SD). A *p*-value < 0.05 was considered significant.

## 3. Results

### 3.1. Construction of Y. pestis Mutants with Lipid A Modification

To exclude potential virulence caused by Pla, *Y. pestis* KIM10 (Pgm^−^ pPCP1^−^) was used as a parent strain to generate mutant strains. Following our previous studies [20,21], we introduced the Δ*lpxP*::P_lpxL_
*lpxL* mutation into *Y. pestis* KIM10 to generate the YPS19 (Δ*lpxP*::P_lpxL_
*lpxL* Pgm^−^ pPCP1^−^) strain (Table 1 and Figure 1), in which *lpxL* genes from *E. coli* encodes the transferase that catalyzes the acyl-oxyacyl linkage of laurate to the 3’ hydroxy-myristate to synthesize hexa-acylated lipid A (Table 1 and Figure 1). Then, the ∆*lacI*::P_lpp_
*lpxE* gene fragment was introduced into YPS19 to replace the *lacI* gene and form the YPS20 (Δ*lpxP*::P_lpxL_
*lpxL* ∆*lacI*::P_lpp_
*lpxE* Pgm^−^ pPCP1^−^) strain (Table 1 and Figure 1), in which the *lpxE* gene encodes the lipid A 1-phosphatase from *F. novicida* that catalyzes removal of the 1-phosphate group from hexa-acylated lipid A to synthesize monophosphoryl lipid A (Table 1 and Figure 1).

### 3.2. Virulence of Y. pestis Mutants in Mice and Protective Immunity 

Previous studies showed that BALB/c mice injected intramuscularly (i.m.) with 1 × 10^7^ CFU of *Y. pestis* KIM D27, a Pgm^−^ strain harboring all three virulence plasmids (pCD1, pMT1, and pPCP1), presented clinical symptoms (ruffled fur and lethargy) and 30% mortality [38]. Therefore, 1 × 10^7^ CFU of KIM10(pCD1Ap), YPS19(pCD1Ap) or YPS20(pCD1Ap) as a minimal dose was administrated by intramuscular (i.m.) or subcutaneous (s.c.) route in this study. Groups of Swiss Webster mice (*n =* 10, equal males and females) were intramuscularly administrated with KIM10(pCD1Ap), YPS19(pCD1Ap) and YPS20(pCD1Ap), respectively and monitored for 30 days to record survivals. Results showed that 60% and 90% of mice survived infection with 5 × 10^7^ CFU and 1 × 10^7^ CFU of KIM10(pCD1Ap) by i.m. injection, respectively; 60% and 100% mice survived infection with 1 × 10^8^ CFU and 1.7 × 10^7^ CFU of YPS19(pCD1Ap), respectively; no mice died of infection with 5 × 10^7^ CFU of YPS20(pCD1Ap) (Figure 2A). The s.c. injection with 3 × 10^7^ CFU of KIM10(pCD1Ap), 1 × 10^8^ CFU of YPS19(pCD1Ap) or 2 × 10^7^ CFU of YPS20(pCD1Ap) showed 80%, 90% or 100% survival, respectively (Figure 2B). No mice succumbed by intranasal (i.n.) instillation with 5 × 10^6^ CFU of KIM10(pCD1Ap), YPS19(pCD1Ap) and YPS20(pCD1Ap), respectively (data not shown). 

On day 42 after administration, the animals that survived from each corresponding group by i.m. administration were pooled together and split in half (*n =* 8, equal males and females) for s.c. or i.n. challenge with virulent *Y. pestis*, respectively. All KIM10(pCD1Ap)- and YPS19(pCD1Ap)-i.m. injected mice survived s.c. challenge with 5 × 10^5^ CFU (50,000 LD_50_) of *Y. pestis* KIM6+(pCD1Ap), while only 20% of the YPS20(pCD1Ap)-i.m. injected mice survived the same challenge (Figure 3A). The i.m. administration with KIM10(pCD1Ap) or YPS19(pCD1Ap) afforded 100% and 88% protection in mice against i.n. challenge with a median dose of 5 × 10^3^ (50 LD_50_) of *Y. pestis* KIM6+(pCD1Ap), respectively, but the i.m. administration with YPS20(pCD1Ap) afforded 20% protection in mice against the pulmonary challenge (Figure 3B). 

In addition, the animals that survived from each corresponding group by s.c. injection on day 42 after administration were pooled together and split in half (*n =* 8, equal males and females) for s.c. or i.n. challenge with virulent *Y. pestis*, respectively. Around 50% of the KIM10(pCD1Ap)-, YPS19(pCD1Ap)- and YPS20(pCD1Ap)-s.c. administrated mice survived s.c. challenge with 50,000 LD_50_ of *Y. pestis* KIM6+(pCD1Ap) (Figure 3C). The s.c. administration with KIM10(pCD1Ap) and YPS19(pCD1Ap) afforded 50% and 40% protection in mice against i.n. challenge with 50 LD_50_ of *Y. pestis* KIM6+(pCD1Ap), respectively. The s.c. administration with YPS20(pCD1Ap) only afforded 20% protection in mice against the pulmonary challenge (Figure 3D). None of the mice administrated PBS survived challenge by either route (Figure 3). 

We analyzed serum IgG responses to *Y. pestis* whole cell lysates (YpL) from i.m. and s.c. administrated mice at week 4 post administration. KIM10(pCD1Ap)-, YPS19(pCD1Ap)- and YPS20(pCD1Ap)-i.m. inoculation primed similar levels of total anti-YpL IgG in mice without substantial difference. In addition, KIM10(pCD1Ap)-, YPS19(pCD1Ap)- and YPS20(pCD1Ap)- s.c. inoculation primed similar levels of total anti-YpL IgG among each group of mice. However, the overall anti-YPL IgG titers in mice by i.m. inoculation were significantly higher than those in mice by s.c. inoculation (Figure 4A). The serum immune responses to YpL were further examined by measuring the levels of IgG isotype subclasses IgG1 and IgG2a. The IgG2a/IgG1 ratio of anti-YpL titers in mice by i.m. administration was close to 1 (Figure 4B), indicating a mixed Th1/Th2 response was stimulated by KIM10(pCD1Ap)-, YPS19(pCD1Ap)- or YPS20(pCD1Ap)-i.m. inoculation. However, the IgG2a/IgG1 ratio of anti-YpL titers in mice by s.c. administration was around 0.5 (Figure 4C), indicating a Th2-biased response was stimulated by KIM10(pCD1Ap)-, YPS19(pCD1Ap)- or YPS20(pCD1Ap)-s.c. inoculation. 

### 3.3. Protection Efficiency Against Pulmonary Y. pestis Infection and Serum Immune Responses 

Regarding the above data, we established that the YPS20(pCD1Ap) strain showed obviously low immunogenicity compared with KIM10(pCD1Ap) or YPS19(pCD1Ap), and the i.m. inoculation with KIM10(pCD1Ap) or YPS19(pCD1Ap) afforded better protection against plague than the s.c. inoculation did and also provided complete protection against s.c. challenge with 50,000 LD_50_ of *Y. pestis* KIM6+(pCD1Ap) (Figure 3A) and significant protection against i.n. challenge with 50 LD_50_ of *Y. pestis* KIM6+(pCD1Ap) (Figure 3B). Therefore, we further evaluated protective efficacy of i.m. immunization with KIM10(pCD1Ap) and YPS19(pCD1Ap) by the prime-boost regime against pulmonary *Y. pestis* challenge. Groups of mice (5 male and 5 female) were immunized with 1 × 10^7^ CFU of KIM10(pCD1Ap) or 2 × 10^7^ CFU of YPS19(pCD1Ap) and boosted at 3 weeks post initial immunization. During immunization, two mice succumbed in the KIM10(pCD1Ap)-immunized group at day 11 post immunization (Figure 5A). Then, all immunized mice were challenged intranasally with 50 LD_50_ of *Y. pestis* KIM6+(pCD1Ap) at 42 days after initial immunization. Both KIM10(pCD1Ap)-immunized and YPS19(pCD1Ap)-immunized groups had the same protective efficacy (survival of 70%). None of the mice administrated PBS survived the pulmonary challenge (Figure 5B). Analysis of serum IgG responses to recombinant LcrV antigen and YpL showed that IgG titers to LcrV or YPL were similar in KIM10(pCD1Ap)- and YPS19(pCD1Ap)-immunized groups at week 2 and displayed increasing trends at week 4 (Figure 5C,D). 

### 3.4. Safety Assessment of Y. pestis Mutant Strains

Virulence assays in Swiss Webster mice showed that virulence of KIM10(pCD1Ap) was slightly higher than that of YPS19(pCD1Ap) by both i.m. and s.c. administration. The attenuated *Y. pestis* Pgm^−^ strain was lethal to an individual with hereditary hemochromatosis manifesting iron overload in tissues [11]. The *Hfe*^−/−^ mice display the iron overload phenotype that was similarly observed in humans with hereditary hemochromatosis [36,39]. Therefore, we used *Hfe*^−/−^ mice to further evaluate safety profiles of KIM10(pCD1Ap) and YPS19(pCD1Ap) strains by i.m. inoculation. The mice (*n =* 10, equal males and females) received 1 × 10^7^ CFU of each strain. Inoculation with YPS19(pCD1Ap) showed 100% survival. However, 80% of *Hfe*^−/−^ mice administrated KIM6+(pCD1Ap) survived by the end of the 30-day observation period. No wild-type C57BL/6 mice died by the same administration with KIM6+(pCD1Ap) or YPS19(pCD1Ap) (Figure 6). The results demonstrated that the YPS19(pCD1Ap) strain as a vaccine candidate had the same protective efficacy as KIM10(pCD1Ap), but seemed safer than KIM10(pCD1Ap) in Swiss Webster and *Hfe*^−/−^ mice.

## 4. Discussion

The success in development of a new generation of effective plague vaccines generally depends on the existence of a suitable vaccine candidate with desirable characteristics capable of eliciting a marked immunity with minimal side effects. This can be accomplished by using technologies to significantly improve both the protection and safety characteristics of the vaccine candidate by modifying particular bacterial properties.

In addition, the outbred Swiss Webster mouse strain has been used as a small-animal model for testing the immunogenicity and efficacy of the rF1V vaccine against plague [40,41,42]. The heterogeneous immune responses expected in Swiss Webster mice due to inherent genetic differences could be a more accurate reflection of the expected human response, rather than in an inbred strain of mouse [40]. Therefore, evaluation of live attenuated *Y. pestis* vaccine candidates in the outbred mice would be more relevant to translation of mouse studies to human application.

*Y. pestis* with a single *lpxL* insertion synthesizing hexa-acylated lipid A induced strong inflammatory responses [17,20]. Therefore, this phenomenon would be analogous to the YPS19(pCD1Ap) strain with the single *lpxL* insertion. Virulence evaluation showed that YPS19(pCD1Ap) clearly reduced its virulence in Swiss Webster mice compared with its parent strain KIM10(pCD1Ap) by i.m. and s.c. administrations, respectively (Figure 2A,B), while the YPS20(pCD1Ap) strain with *lpxL* and *lpxE* double insertions synthesizing monophosphoryl lipid A showed complete attenuation (Figure 2A,B). However, our previous study showed that χ10027(pCD1Ap) (∆*lpxP*::P_lpxL_
*lpxL* ∆*lacI*::P_lpp_
*lpxE*) with both *lpxL* and *lpxE* insertions derived from the KIM6+(pCD1Ap) (Pgm^+^ pPCP1^+^) strain was more virulent than χ10015(pCD1Ap) (∆*lpxP*::P_lpxL_
*lpxL*), even though strain χ10027(pCD1Ap) significantly decreased Pla activity and slightly affected Pla synthesis compared to χ10015(pCD1Ap) and KIM6+(pCD1Ap) [21]. Removal of Pla, a virulence factor encoded on the pPCP1 plasmid, clearly reduced the virulence of *Y. pestis* [25,27,30]. Therefore, lipid A 1-dephosphorylation may have pleiotropic effects on other virulence factors in *Y. pestis*. In addition, we observed that strain χ10027(pCD1Ap) produced more biofilm than χ10015(pCD1Ap) and KIM6+(pCD1Ap) in the same culture conditions (observation data). In *Y. pestis*, biofilm formation is primarily dependent upon genes of the *hms* system, including the *hmsHFRS* genes within the *pgm* (pigmentation) locus [43,44]. Thus, the lipid A 1-dephosphorylation in *Y. pestis* might upregulate genes such as *psn* [45], *ripA* [33] and others along with *hmsHFRS* within the *pgm* locus to increase bacterial resistance in mammalian hosts. Both YPS19(pCD1Ap) and YPS20(pCD1Ap) were derived from KIM10(pCD1Ap) and lack pPCP1 and the *pgm* locus, so lipid A 1-dephosphorylation in YPS20(pCD1Ap) loses this ability to upregulate virulence factors within the *pgm* locus, which may be one of reasons why YPS20(pCD1Ap) shows higher attenuation than YPS19(pCD1Ap).

Intramuscularly administrated YPS20(pCD1Ap) could not generate effective protection against virulent *Y. pestis*(pCD1Ap) challenge as immunization with KIM10(pCD1Ap) or YPS19(pCD1Ap) did (Figure 3A,B), although administration of YPS20(pCD1Ap) primed similar levels of anti-*Yersinia* IgG response to administration of KIM10(pCD1Ap) or YPS19(pCD1Ap) (Figure 4A). It is worthy noticing that the residual virulence in the live *Y. pestis* EV vaccine strain is considered to be necessary for the development of adequate immunity against plague [3,46]. Previous studies showed that vaccination with attenuated Pgm^−^ derived *Y. pestis* strains primed T cells in an antibody-independent system that protect mice against pneumonic plague [47,48]. Thus, another reason for this explanation is that antibody responses induced by immunization with the above *Y. pestis* mutants are not essential for protection. We also noticed that i.m. immunization with attenuated *Y. pestis* mutants generated balanced Th1/Th2 responses, while s.c. immunization with corresponding strains stimulated Th2-biased responses (Figure 4B,C), which can explain why mice who survived s.c. administration were conferred less protection against virulent *Y. pestis* KIM6+(pCD1Ap) challenge than i.m. administration, except for the YPS20(pCD1Ap) strain (Figure 3).

Mice vaccinated i.m with KIM10(pCD1Ap) by a prime-boost strategy still had two deaths without significance compared to YPS19(pCD1Ap) (Figure 5A). Immunization with each strain in Swiss Webster mice conferred significant protection (70%) against pulmonary *Y. pestis* challenge without further enhancement (Figure 5B). The variations of protective efficacy compared to Figure 3B may be due to genetic variations in those outbred mice. Further, safety evaluation suggested that YPS19(pCD1Ap) was safer than KIM10(pCD1Ap) in *Hfe*^−/−^ mice. Overall, the advantageous features of the YPS19(pCD1Ap) (Δ*lpxP*::P_lpxL_
*lpxL* Pgm^−^ pPCP1^−^) makes it a potential candidate for an improved live plague vaccine because it is high attenuation without weakening its protection. This study is paving a way to develop a safe, live *Y. pestis* vaccine for counteracting human plague. 

## 5. Conclusions

Virulence evaluation in mice demonstrated that YPS19(pCD1Ap) mainly synthesizing the hexa-acylated lipid A was further attenuated in comparison to its parent strain, KIM10(pCD1Ap) by i.m. or s.c. administration. The i.m. immunization with either KIM10(pCD1Ap) or YPS19(pCD1Ap) strain induced similar levels of antibody response manifesting a balanced Th1/Th2 response and afforded the same protective efficacy against *Y. pestis* infection. Therefore, YPS19(pCD1Ap) would be a potential vaccine candidate with improved features.

## Figures and Tables

**Figure 1 vaccines-08-00095-f001:**
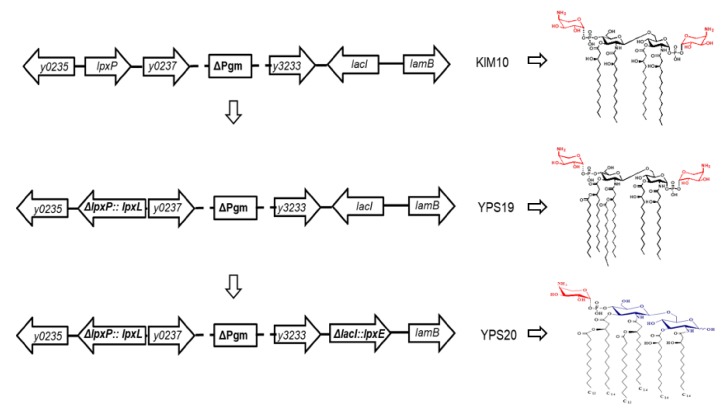
Schematic chromosome structure of *Y. pestis* KIM10 (Pgm^−^ pPCP1^−^, pCD1^−^), YPS19 (Δ*lpxP32*::P_lpxL_
*lpxL*, Pgm^−^ pPCP1^−^, pCD1^−^), and YPS20 (Δ*lpxP32*::P_lpxL_
*lpxL* ∆*lacI23*::P_lpp_
*lpxE*, Pgm^−^ pPCP1^−^, pCD1^−^) and their corresponding lipid A structures based on our previous reports [20,21].

**Figure 2 vaccines-08-00095-f002:**
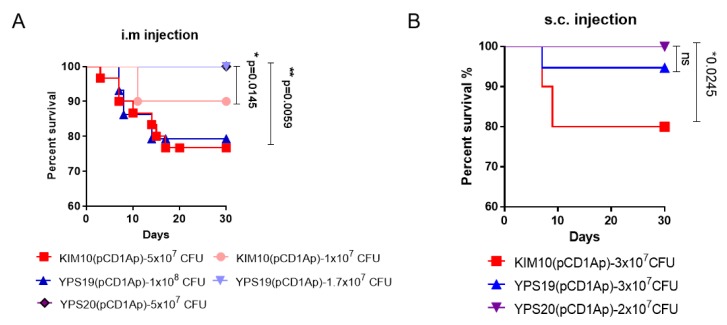
Survival of Swiss Webster mice administrated i.m. or s.c. with *Y. pestis* KIM10(pCD1Ap), YPS19(pCD1Ap), YPS20(pCD1Ap), respectively. (**A**) Mice (*n =* 10, equal males and females) were administrated i.m. with 5 × 10^7^ CFU or 1 × 10^7^ CFU of KIM10(pCD1Ap), with 1 × 10^8^ CFU or 1.7 × 10^7^ CFU of YPS19(pCD1Ap), and 5 × 10^7^ CFU of YPS20(pCD1Ap), respectively; (**B**) Mice (*n =* 10, equal males and females) were administrated s.c. with 3 × 10^7^ CFU of KIM10(pCD1Ap), with 3 × 10^7^ CFU of YPS19(pCD1Ap), and 2 × 10^7^ CFU of YPS20(pCD1Ap), respectively. The log-rank (Mantel–Cox) test was used for statistical analysis, *****, the *p* value was less than 0.05; **, the *p* value was less than 0.005; ns, no significance.

**Figure 3 vaccines-08-00095-f003:**
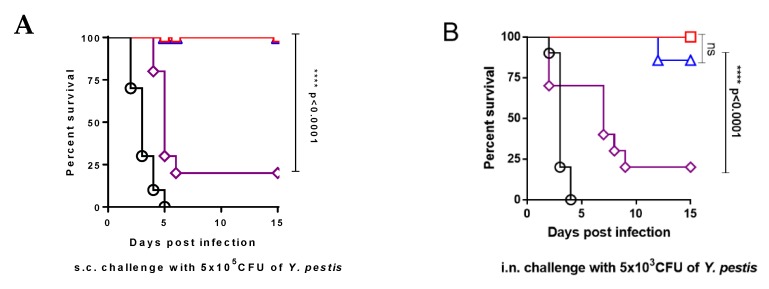
Mouse survival after *Y. pestis* KIM6+(pCD1Ap) challenge. Mice that survived after i.m. or s.c. administration were pooled and split evenly for s.c. and i.n. challenge of virulent *Y. pestis* KIM6+(pCD1Ap), respectively. (**A**) Swiss Webster mice administrated i.m. (*n =* 8) were challenged with 5 × 10^5^ CFU of *Y. pestis* KIM6+(pCD1Ap) via the s.c. route. (**B**) Swiss Webster mice administrated i.m. (*n =* 8) were challenged via the i.n. route with 5 × 10^3^ CFU of *Y. pestis* KIM6+(pCD1Ap). (**C**) Swiss Webster mice administrated s.c. (*n =* 8) were challenged with 5 × 10^5^ CFU of *Y. pestis* KIM6+(pCD1Ap) via the s.c. route. (**D**) Swiss Webster mice administrated s.c. (*n =* 8) were challenged via the i.n. route with 5 × 10^3^ CFU of *Y. pestis* KIM6+(pCD1Ap). For each experiment, 5 mice administrated with PBS served as negative controls. The log-rank (Mantel–Cox) test was used for statistical analysis, ********, the *p* value was less than 0.0001; ns, no significance.

**Figure 4 vaccines-08-00095-f004:**
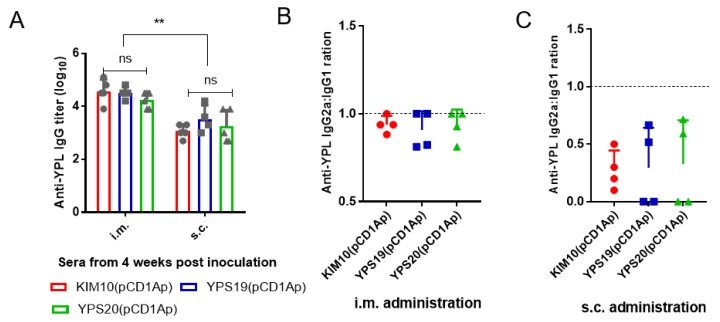
Antibody response in sera of mice administrated i.m. or s.c. with *Y. pestis* KIM10(pCD1Ap), YPS19(pCD1Ap), YPS20(pCD1Ap), respectively at week 4 post administration. A *Y. pestis* whole cell lysate (YPL) was used as the coating antigen. (**A**) Serum IgG responses in mice by i.m. or s.c. administration; (**B**) Ratio of serum IgG2a and IgG1 responses in mice by i.m. administration; (**C**) Ratio of serum IgG2a and IgG1 responses in mice by s.c. administration. The two-way ANOVA, ******, the *p* value was less than 0.05.

**Figure 5 vaccines-08-00095-f005:**
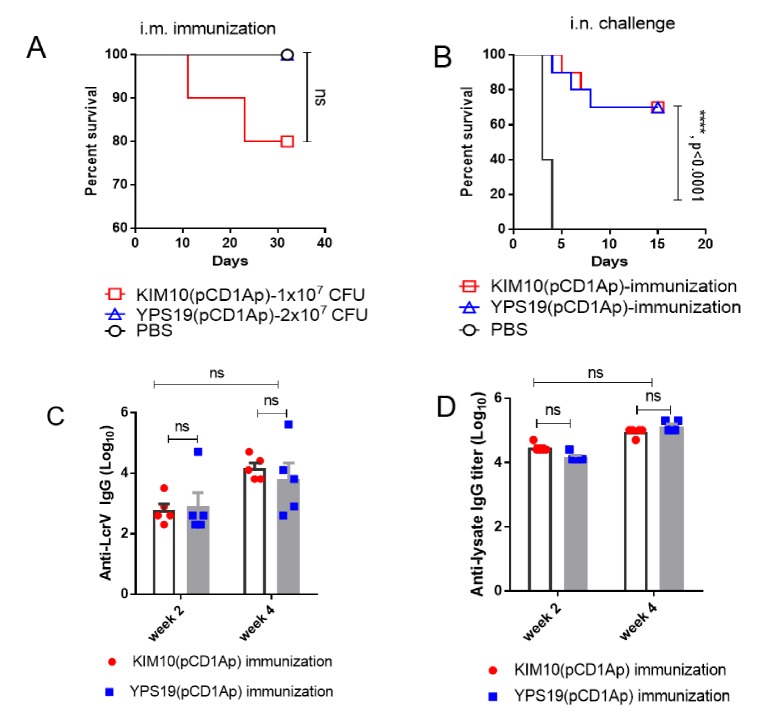
Mouse survival after pulmonary *Y. pestis* KIM6+(pCD1Ap) challenge and serum response. Mice (*n =* 10, equal males and females) were vaccinated i.m. with 1 × 10^7^ CFU of KIM10(pCD1Ap), and 2 × 10^7^ CFU of YPS19(pCD1Ap), respectively, and then boosted at 21 days post vaccination with corresponding strains. On day 42 after initial vaccination, the vaccinated mice were challenged i.n. with virulent *Y. pestis* KIM6+(pCD1Ap). (**A**) Survival of mice after i.m. vaccination with KIM10(pCD1Ap) and YPS19(pCD1Ap), respectively. (**B**) Vaccinated Swiss Webster mice were challenged via the i.n. route with 5 × 10^3^ CFU of *Y. pestis* KIM6+(pCD1Ap). The log-rank (Mantel–Cox) test was used for statistical analysis, ********, the *p* value was less than 0.0001. (**C**) Serum IgG titers to LcrV antigen. (**D**) Serum IgG titers to *Y. pestis* whole cell lysate (YPL). The two-way ANOVA, ns, no significance.

**Figure 6 vaccines-08-00095-f006:**
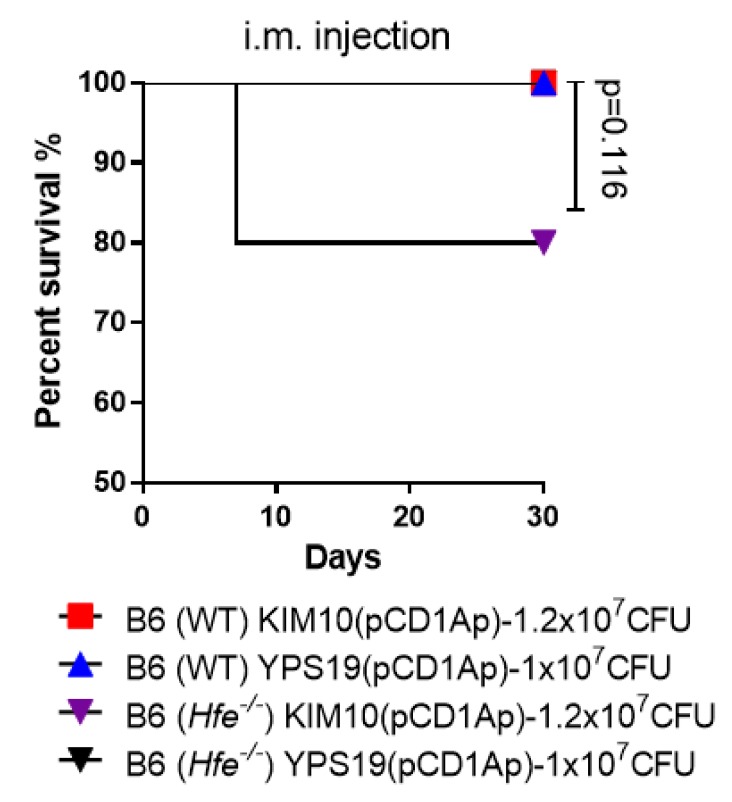
Safety assessment of live attenuated *Y. pestis* vaccine strains in wild-type and *Hfe^−/−^* C57BL/6 mice. A single dose of 10^7^ CFU of KIM10(pCD1Ap) and YPS19(pCD1Ap) were administrated i.m. to wild-type and *Hfe^−/−^* C57BL/6 mice (*n =* 10, 5 males and 5 females), respectively. These mice were observed for signs of mortality and morbidity for 30 days. The log-rank (Mantel–Cox) test was used for statistical analysis, the *p* value is 0.116.

**Table 1 vaccines-08-00095-t001:** Bacterial strains and plasmids used in this study.

Strain	Relevant Genotype or Annotation	Source
*E. coli* TOP10	F^−^ *mcr**A* ∆(*mrr*-*hsd**RMS*-*mcr**BC*) φ80*lac*Z∆*M15* ∆*lacX74 rec**A1 ara**D139* ∆(*ara*-*leu*)*7697 gal**U galK rpsL endA1 nupG*	Invitrogen
*Y. pestis* KIM6+(pCD1Ap)	Pgm^+^, pMT1, pPCP1, pCD1Ap	(1)
*Y. pestis* KIM10	Pgm^−^, pMT1, pPCP1^−^	[33]
YPS19	∆*lpxP32*::P_lpxL_ *lpxL* Pgm^−^ pPCP1^−^	This study
YPS20	∆*lpxP32*::P_lpxL_ *lpxL* ∆*lacI23*::P_lpp_ *lpxE* Pgm^−^ pPCP1^−^	This study
KIM10(pCD1Ap)	pCD1Ap, pPCP1^−^, Pgm^−^	This study
YPS19(pCD1Ap)	pCD1Ap, pPCP1^−^, Pgm^−^, ∆*lpxP32*::P_lpxL_ *lpxL*	This study
YPS20(pCD1Ap)	pCD1Ap, pPCP1^−^, Pgm^−^, ∆*lpxP32*::P_lpxL_ *lpxL* ∆*lacI23*::P_lpp_ *lpxE*	This study
Plasmid		Source
pKD46	λ Red recombinase expression plasmid	[34]
pYA4373	The *cat*-*sacB* cassette in the *Pst*I and *Sac*I sites of pUC18.	[35]
pYA4577	The P_lpxL_*lpxL* gene fragment flanked by the *lpxP* upstream and downstream sequence	[20]
pYA4578	The *cat*-*sacB-*P_lpxL_*lpxL* gene fragment flanked by the *lpxP* upstream and downstream sequence	[20]
pYA4735	The P_lpp_ *lpxE* gene fragment flanked by the *lacI* upstream and downstream sequence	[21]
pYA4736	The *cat*-*sacB-*P_lpp_ *lpxE* gene fragment flanked by the *lacI* upstream and downstream sequence	[21]

**Table 2 vaccines-08-00095-t002:** Primers used in this work.

Name	Sequence
LpxL1	5′gggagctccgctgatttgcgcgttaatgccctca3′
LpxL2	5′cggctgcaggaacataagaagaaaagataag3′
LpxE1	5′cgggagctcggataaccagaagcaataaaaaatc3′
LpxE2	5′cgggagctcctaaataatctcacgattacgca3′
Cm-V	5′gttgtccatattggccacgttta3′
SacB-V	5′gcagaagagatatttttaattgtgga3′
pPCP1^−^V1	5′cgggaattcagcaaaacagacaaacgcctgctgg3′
pPCP1^−^V2	5′cggctgcagtagacacccttaatctctctgcatg3′

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
