# Peer review of "Protection and Safety Evaluation of Live Constructions Derived from the Pgm? and pPCP1? Yersinia pestis Strain"

_vaccines, 2020, doi:10.3390/vaccines8010095_

Round 1

Reviewer 1 Report

This is a good conceptual manuscript, but I have a few suggestions/recommendations, below;

Line 46-48: briefly explain how Y.pestis strain infect and evades the host system, and fix the grammatical error, "...in the mammalian due to..". Mammalian what?

Line 99: Grammatical error: "... was followed our previous...". I would recommend briefly explaining how the mutant strains are constructed, as this manuscript is focused on various strains. Asking readers to refer to another publication for the core concepts is not prudent.

Line 109 & 114: Does 100uL of bacterial suspension in line 109 relate to the 10E7 CFU in 100uL Saline in line 114?

Line 130: please re-word the beginning part of the sentence.

Line 153: please clarify that KIM10, YPS19, YPS20 were administered by just one route, i.e. IM, SC, & IN? as the subsequent parts in the section state otherwise, as stated on line 157. This section is confusing due to way it is written.

Line 166: Pertaining to the explanation about Fig. 3A, while the actual fig. 3A is missing the Yp19 strain representation (blue line) in the plot.

Line 163 & 172: please change the word pool to 'pooled', as the manuscript is written is past tense.

Author Response

This is a good conceptual manuscript, but I have a few suggestions/recommendations, below;

Line 46-48: briefly explain how Y.pestis strain infect and evades the host system, and fix the grammatical error, "...in the mammalian due to..". Mammalian what?

Response: Thanks for your comments and pointing out this error. We made revisions in the updated manuscript. Page 2 line 43.

Line 99: Grammatical error: "... was followed our previous...". I would recommend briefly explaining how the mutant strains are constructed, as this manuscript is focused on various strains. Asking readers to refer to another publication for the core concepts is not prudent.

Response: Thanks for your comments and pointing out this error. We added the procedure for strain construction in section 2.3 (page 3 line 96-110)

Line 109 & 114: Does 100uL of bacterial suspension in line 109 relate to the 10E7 CFU in 100uL Saline in line 114?

Response: Thanks for your comments. We added exact colony-forming unit in page 3 line 120-121.

Line 130: please re-word the beginning part of the sentence.

Response: Thanks for pointing out this error. We corrected it.

Line 153: please clarify that KIM10, YPS19, YPS20 were administered by just one route, i.e. IM, SC, & IN? as the subsequent parts in the section state otherwise, as stated on line 157. This section is confusing due to way it is written.

Response: Thanks for your comments. We rewrote this part (Page4 line 162-172).

Line 166: Pertaining to the explanation about Fig. 3A, while the actual fig. 3A is missing the Yp19 strain representation (blue line) in the plot.

Response: Thanks for your comments. The YPS19 strain was shown in the Fig. 3A (Page 14).

Line 163 & 172: please change the word pool to 'pooled', as the manuscript is written is past tense.

Response: Thanks for pointing out this error. We corrected it.

Reviewer 2 Report

Authors reported on the modification of Yersinia pestis vaccine by its specific mutation. Research has been prepared reliably. Applied methodology has been designed adequately, investigations have been described properly and they were followed by extensive discussion on the results obtained. Paper is worth considering for publication due to its high scientific value but some minor improvements are needed. All suggestions are listed below.

  • Abstract of the paper needs to be re-written. It is full of specialized nomenclature and contains numerous abbreviations that make it difficult to understand the content. Therefore it should be written in a more accessible and understandable way. Additionally, the novelty of the research needs to be clearly emphasized.
  • Section Introduction: Authors mentioned that: “The EV76 vaccine (…) sometimes caused local and systemic reactions”. What type were these reaction ? Please develop this issue in a more detail.
  • Procedure of the insertion of selected genes into Yersinia pestis (section 2.3.) should be described by Authors in few sentences.
  • It will be better to summarize obtained results in separate section defined as “Conclusions”. Extensive discussion presented by Authors is proper and really impressive but I suggest to contain the essential highlights of the research separately to make manuscript more readable.
  • Some References should be corrected. As required by the journal, all references should contain abbreviated journal titles. In some cases – i.e. 23 or 42 – the whole titles are presented.

Author Response

Authors reported on the modification of Yersinia pestis vaccine by its specific mutation. Research has been prepared reliably. Applied methodology has been designed adequately, investigations have been described properly and they were followed by extensive discussion on the results obtained. Paper is worth considering for publication due to its high scientific value but some minor improvements are needed.

Response: Thanks for your comments!

All suggestions are listed below.

  • Abstract of the paper needs to be re-written. It is full of specialized nomenclature and contains numerous abbreviations that make it difficult to understand the content. Therefore it should be written in a more accessible and understandable way. Additionally, the novelty of the research needs to be clearly emphasized.

Response: Thanks for your comments. The abstract was rewritten completely in an understandalbe way.

  • Section Introduction: Authors mentioned that: “The EV76 vaccine (…) sometimes caused local and systemic reactions”. What type were these reaction ? Please develop this issue in a more detail.

Response: Thanks for your comments and pointing out this error. We made revisions in the updated manuscript. Page 1 line 31-32.

  • Procedure of the insertion of selected genes into Yersinia pestis (section 2.3.) should be described by Authors in few sentences.

Response: Thanks for your comments and pointing out this error. We added the procedure for strain construction in section 2.3 (page 3 line 96-110)

  • It will be better to summarize obtained results in separate section defined as “Conclusions”. Extensive discussion presented by Authors is proper and really impressive but I suggest to contain the essential highlights of the research separately to make manuscript more readable.

Response: Thanks for your comments. We added the “conclusion” part at the end discussion (page 7 line 293-298).